# Conic Descent and its Application to Memory-efficient Optimization over Positive Semidefinite Matrices

**John Duchi**
Stanford University
jduchi@stanford.edu

**Oliver Hinder**
Stanford University, Google, University of Pittsburgh
ohinder@pitt.edu

**Andrew Naber**
Stanford University
naber@stanford.edu

**Yinyu Ye**
Stanford University
yyye@stanford.edu

## Abstract

We present an extension of the conditional gradient method to problems whose feasible sets are convex cones. We provide a convergence analysis for the method and for variants with nonconvex objectives, and we extend the analysis to practical cases with effective line search strategies. For the specific case of the positive semidefinite cone, we present a memory-efficient version based on randomized matrix sketches and advocate a heuristic greedy step that greatly improves its practical performance. Numerical results on phase retrieval and matrix completion problems indicate that our method can offer substantial advantages over traditional conditional gradient and Burer-Monteiro approaches.

## 1 Introduction

We want to solve problems of the form

$$
\begin{aligned}
\text{minimize} \quad & f(x) \\
\text{subject to} \quad & x \in K
\end{aligned}
\tag{1}
$$

where $K \subset \mathbf{R}^n$ is a proper, convex cone and $f$ is a convex, differentiable function which has no nonzero direction of recession in $K$, *i.e.*, it eventually curves upward along any nonzero ray in $K$. Let $p^\star$ denote its optimal value. These assumptions imply that an optimal solution $x^\star$ exists. Our work shows that it is possible to directly solve such a problem using a modification of the conditional gradient algorithm that we call conic descent (CD). At each iteration, CD picks a descent direction which is a conic combination of a step toward the origin and a direction in $K$. If CD fails to find a descent direction, then an optimality certificate can be found.

We can add a redundant constraint $\|x\| \leq R$ to problem (1) as long as $\|x^\star\| \leq R$. This ensures that the feasible set is compact, and the well-known conditional gradient algorithm could be applied. With a slight abuse of terminology, we will hereafter refer to this approach as simply CG. Unfortunately, $R$ is usually unknown in applications, and overestimates can greatly reduce the performance of CG. Appendix E provides a worked example of this.

**Prior work on projection-free optimization over cones** In [12], the authors propose a version of CG which is similar to CD in that it finds feasible directions of descent from a conic combination of a step toward the origin and a direction in $K$; however, in distinction with CD, the method requires an

upper bound on $\|x^\star\|$ in order to solve the subproblems. We also obtain a small (factor of 4) constant improvement over their rate of convergence.

Optimization of differentiable functions over conic hulls is addressed in [17] which builds off earlier work in [22]. The basic method they propose obtains the same $O(1/k)$ convergence bound as CD; however, our method and analysis improves the constants by removing a quadratic dependency on the geometry of the set used to define the cone (compare Theorem 2 in [17] with our Theorem 1). Our CD method is more general in that it can work for an arbitrary cone—even if there is no known simple description of $K$ as a conic hull (*e.g.*, the doubly nonnegative cone). Furthermore, our proof of convergence is simple and based on convex duality.

**Memory and the positive semidefinite cone** Optimization over the positive semidefinite cone is very important in statistics and machine learning. Both phase retrieval and matrix completion have natural formulations as such problems (for example, see [6, 7]). Motivated by these applications, we are interested in problems for which $n$ is very large. There have been CG-like algorithms designed specifically for this case; see [11, 16] for example. However, these methods do not directly address the $O(n^2)$ memory consumption of the full matrix variable which is frequently a bottleneck.

To directly address the memory requirement, it is common to employ the Burer-Monteiro factorization heuristic (named after the authors of its initial proposal and analysis in [4, 5]). This heuristic has excellent empirical performance and has inspired a lot of analysis. Under various strong assumptions on $f$, it can sometimes be shown that the Burer-Monteiro heuristic does indeed give solutions to problem (1). For example, see the related work in [2, 9, 13, 19, 20]. However, it can be very difficult to verify such assumptions in practical situations.

A novel method of dealing with the memory requirement was presented in [24]. In this paper, the authors exploited the fact that applying CG to problems with feasible set $\{X \mid X \in \mathbf{S}_+^n, \mathbf{tr}\, X \le d\}$ for $d \ge 0$ generates the optimal solution as a sum of rank-1 matrices. They showed how an auxiliary variable could be used to drive the iterations of CG and how a memory-efficient randomized sketch of the iterate $X_k$ could be maintained throughout the iterations. Upon convergence, the sketch could be used to form an approximation to the optimal solution. Assuming that the solution to which CG converged had low rank, this method can recover the exact solution without ever forming the full matrix variable. This approach is very promising because it only makes weak assumptions on the nature of the objective and directly solves problem (1). However, the trace constraint on the feasible set is limiting.

Because CD also generates solutions to problems over the positive semidefinite cone as a sum of rank-1 matrices, we propose a memory-efficient version of CD in the spirit of [24] that allows us to track the iterate using randomized sketches. In addition, the nature of CD allows us to easily incorporate greedy steps using a Burer-Monteiro factorization heuristic. In effect, we can get the best of both approaches: global convergence guarantees using low memory without any prior bound on $\mathbf{tr}(X^\star)$ and rapid convergence when the factorization heuristic is working well.

**Contributions**

- We introduce conic descent as a method for solving (1) that, unlike prior work, requires no bound on $\|x^\star\|$ or description of $K$ as a conic hull. Our proof of convergence is simple and based on convex duality. Our analysis of the rate of convergence does not depend on any constants relating to the specific cone.

- We provide analyses of several natural extensions to CD, including the use of a backtracking line search to determine the step size as well as the use of inexact methods for rescaling and solving the subproblem at each iteration. Additionally, we show that CD applied to a nonconvex objective converges to a stationary point and provide an analysis of its rate of convergence.

- We provide a memory-efficient modification to CD for the positive semidefinite cone based on randomized matrix sketches that is similar to [24] but does not require any prior bound on $\|X^\star\|$. Additionally, we propose a greedy factorization heuristic that adds excellent practical performance while retaining the convergence guarantees of CD.

- We provide a Python implementation of the memory-efficient modification to CD with the greedy heuristic. It is available at `https://github.com/AndrewTNaber/memeffpsd`.

**Notation**   The $n$-dimensional nonnegative orthant cone and positive semidefinite cone are denoted $\mathbf{R}_+^n$ and $\mathbf{S}_+^n$, respectively. The notation $\|\cdot\|$ denotes an arbitrary norm with dual norm $\|x\|_* = \sup\{y^T x \mid \|y\| \le 1\}$. The vector of all ones is denoted $\mathbf{1}$; its dimension will be clear from context. The dual cone of $K$ is denoted $K^*$. The distance between a point $x$ and a set $Y$ with respect to the dual norm of some arbitrary norm is denoted $\mathbf{dist}^*(x, Y)$. The minimum eigenvalue of a symmetric matrix $X$ is denoted $\lambda_{\min}(X)$. For a matrix $X$, its pseudoinverse is denoted with $X^\dagger$, and its $i$th singular value is denoted $\sigma_i(X)$.

## 2   Algorithm

Because $f$ is convex and $K$ is proper, Slater's condition guarantees that the Karush-Kuhn-Tucker (KKT) conditions are necessary and sufficient for optimality in problem (1). Let $\epsilon > 0$ be a tolerance. A point $x_\epsilon^\star$ is said to $\epsilon$-approximately satisfy the KKT conditions if

$$x_\epsilon^\star \in K, \quad \nabla f(x_\epsilon^\star)^T x_\epsilon^\star = 0, \quad \text{and} \quad \mathbf{dist}^*(\nabla f(x_\epsilon^\star), K^*) \le \epsilon. \tag{2}$$

We propose Algorithm 1 for finding such a point.

---
**Algorithm 1** Conic descent

---
1: $x_0 = 0$
2: **for** $k = 1, 2, 3, \ldots$ **do**
3:     find $\eta_k \in \operatorname{argmin}_{\eta \ge 0} f(\eta x_k)$
4:     find $v_k \in \operatorname{argmin}\{\nabla f(\eta_k x_k)^T v \mid \|v\| \le 1, v \in K\}$
5:     **if** $\nabla f(\eta_k x_k)^T v_k > -\epsilon$ **then**
6:         **break** with $x_\epsilon^\star = \eta_k x_k$
7:     **else**
8:         find $\theta_k \in \operatorname{argmin}_{\theta \ge 0} f(\eta_k x_k + \theta v_k)$
9:         $x_{k+1} = \eta_k x_k + \theta_k v_k$
10:     **end if**
11:     (optional greedy step)
12: **end for**

---

We now discuss the correctness of this method. First, the iterates it generates are always feasible. This can be seen inductively. Trivially, $x_0 \in K$. Now suppose that $x_k \in K$, then $x_{k+1} \in K$ because $\eta_k$ and $\theta_k$ are nonnegative and $v_k \in K$. Second, the rescaling in step 3 ensures that the complementary slackness holds for $\eta_k x_k$. Because $f$ has no nonzero direction of recession in $K$ (by assumption), $\eta_k$ must be finite. Thus, $\nabla f(\eta_k x_k)^T (\eta_k x_k) = 0$ because $\eta_k = 0$ or $\eta_k$ satisfies $\frac{d}{d\eta} f(\eta x_k) = \nabla f(\eta x_k)^T x_k = 0$. Finally, a dual of the subproblem in line 4 is given by

$$\begin{aligned} \text{maximize} \quad & -\|\nabla f(\eta_k x_k) - u\|_* \\ \text{subject to} \quad & u \in K^* \end{aligned}$$

This dual problem is simply to find the projection of $\nabla f(\eta_k x_k)$ onto $K^*$ and its optimal value is $-\mathbf{dist}^*(\nabla f(\eta_k x_k), K^*)$. Strong duality between the subproblem and its dual holds. If $\mathbf{dist}^*(\nabla f(\eta_k x_k), K^*) > \epsilon$, then $\nabla f(\eta_k x_k)^T v_k < -\epsilon$. This implies that that $v_k$ must be a descent direction in $K$. Otherwise, the algorithm terminates with a point $x_\epsilon^\star$ which $\epsilon$-approximately satisfies the KKT conditions (*i.e.*, satisfies (2)).

The optional greedy step in Algorithm 1 is simply a routine which updates $x_{k+1}$ to any feasible point with a lower objective value. It allows us to take advantage of speedy or memory-efficient heuristics for reducing the objective over $K$ while still maintaining the convergence guarantees of CD. The use of a well-chosen greedy step can greatly improve performance. In Section 4, we will elaborate on the use of the Burer-Monteiro factorization heuristic to improve CD when $K = \mathbf{S}_+^n$.

## 3   Theory

### 3.1   Lipschitz continuous gradient (convex objective)

Our analysis of CD is summarized in Theorem 1 and is similar to that of CG; see Theorem 3.8 in [3], for example. The proof can be found in Appendix A. Our method obtains the same $O(1/k)$ rate of

convergence as CG, and it is interesting to compare the constants appearing in both cases. For CG, the constant is proportional to $R^2$ whereas here it only depends on $\|x^\star\|^2$.

**Theorem 1.** *If $f$ has a Lipschitz continuous gradient with respect to $\|\cdot\|$ with parameter L, then conic descent generates feasible points $x_k$ such that $f(x_k) - f(x^\star) \leq \frac{2L\|x^\star\|^2}{k+2}$.*

**Backtracking line search variant** The proof of Theorem 1 shows that three methods of choosing the step size at each iteration obtain the same $O(1/k)$ rate of convergence. If we somehow know $\|x^\star\|$, then we can use $\theta_k = \frac{2\|x^\star\|}{k+2}$. The proof also showed that using an exact line search on $f(\eta_k x_k + \theta v_k)$ for $\theta \geq 0$ to find $\theta_k$ or choosing $\theta_k = -\frac{1}{L}\nabla f(\eta_k x_k)$ achieve the same rate of convergence. The exact line search can be a bit burdensome, and the other two choices rely on constants that are generally unknown. Instead, we can use a backtracking line search to determine $\theta_k$ and obtain the same rate of convergence as these other choices. See Theorem 2; our proof is similar to that of Theorem 1 and can be found in Appendix B.

**Theorem 2.** *If $f$ has a Lipschitz continuous gradient with respect to $\|\cdot\|$ with parameter L, then conic descent with $\theta_k$ determined from a backtracking line search (Algorithm 3) generates feasible points $x_k$ such that $f(x_k) - f(x^\star) \leq \frac{2L\|x^\star\|^2}{\min\{2L\alpha, 4\alpha(1-\alpha)\beta\}(k+2)}$.*

**Inexact minimization variant** Suppose that the rescaling in step 3 of Algorithm 1 is only solved approximately. That is, we obtain an $\eta_k$ such that $f(\eta_k x_k) \leq f(x_k)$ and

$$\nabla f(\eta_k x_k)^T(\eta_k x_k) \leq \frac{\gamma}{k+2}, \tag{3}$$

where $\gamma > 0$. Additionally, suppose that the linear subproblem in step 4 of Algorithm 1 can be solved approximately. Specifically, at each iteration, we obtain a $\tilde{v}_k$ such that

$$\nabla f(\eta_k x_k)^T \tilde{v}_k \leq \nabla f(\eta_k x_k)^T v_k + \frac{\delta}{k+2}, \tag{4}$$

where $v_k$ is the optimal subproblem solution and $\delta > 0$. Under these assumptions, the proof of convergence proceeds similarly to that of Theorem 1. Note that the rescaling and subproblem must be solved with increasing accuracy. Theorem 3 guarantees the same $O(1/k)$ convergence, and its proof can be found in Appendix C.

**Theorem 3.** *If $f$ has a Lipschitz continuous gradient with respect to $\|\cdot\|$ with parameter L, then conic descent with an inexact rescaling method that satisfies (3) and with an an inexact subproblem solution method that satisfies (4) generates feasible points $x_k$ such that $f(x_k) - f(x^\star) \leq \frac{2}{k+2}(\gamma + \delta\|x^\star\| + L\|x^\star\|^2)$.*

### 3.2 Lipschitz continuous gradient (nonconvex objective)

We also consider the case where the objective function may be nonconvex but still has a Lipschitz continuous gradient. The only necessary modification to conic descent is that $\eta_k$ is no longer required to be an exact minimizer of $f(\eta x_k)$ over $\eta \geq 0$ in step 3. We instead require first that $\eta_k = 0$ or $\eta_k$ satisfies $\frac{d}{d\eta}f(\eta x) = \nabla f(\eta x_k)^T x_k = 0$ and second that $f(\eta_k x_k) \leq f(x_k)$. This can be readily achieved using standard univariate optimization tools. With this modification, conic descent still converges to a point which $\epsilon$-approximately satisfies the KKT conditions (*i.e.*, a point which satisfies (2)). However, recall that because $f$ is nonconvex, the KKT conditions are no longer sufficient to guarantee optimality. Our proof can be found in Appendix D and is similar in spirit to that presented in [15] for the conditional gradient method with an objective which is nonconvex but has a Lipschitz continuous gradient, and we obtain the same $O(1/\sqrt{k})$ rate of convergence.

**Theorem 4.** *If $f$ is nonconvex and has a Lipschitz continuous gradient with respect to $\|\cdot\|$ with parameter L, then conic descent with $\theta_k = -\frac{1}{L}\nabla f(\eta_k x_k)^T v_k$ will generate a point which satisfies (2) in at most $\lceil 2L(f(x_0) - f(x^\star))/\epsilon^2 \rceil$ iterations.*

## 4  Memory-efficient optimization over the positive semidefinite cone

We now focus exclusively on the case of the positive semidefinite cone. Specifically, we are interested in solving problems of the form

$$\begin{array}{ll} \text{minimize} & f(\mathcal{G}(X) - g) \\ \text{subject to} & X \in \mathbf{S}_+^n, \end{array} \tag{5}$$

where $g \in \mathbf{R}^m$ and $\mathcal{G}(X) = (\mathbf{tr}(G_1 X), \dots, \mathbf{tr}(G_m X))$ with $G_i \in \mathbf{S}^n$ for $i = 1, \dots, m$. The adjoint of the linear operator $\mathcal{G}$ acting on some $z \in \mathbf{R}^m$ is given by $\mathcal{G}^*(z) = z_1 G_1 + \cdots + z_m G_m$. We make the additional assumptions that $m$ is much smaller than $n^2$ and that there are efficient methods for evaluating matrix-vector products for each $G_i$. Such an assumption implies that it is possible to efficiently compute matrix-vector products for the matrix $\mathcal{G}^*(z)$ as well as efficiently evaluate $\mathcal{G}(UU^T)$ for $U \in \mathbf{R}^{n \times r}$ for $r$ much smaller than $n$. This type of problem appears frequently in statistics and machine learning, where the $G_i$ are often low rank, sparse, or discrete Fourier transform matrices. Under these assumptions, this problem can be solved with limited memory; see Algorithm 2. The bracketed expressions represent optional aspects of the algorithm and will be explained shortly.

---

**Algorithm 2** Memory-efficient conic descent for problem (5)

---

1: $\langle$form $\Omega \in \mathbf{R}^{n \times r}$ with entries independently drawn from a standard normal distribution$\rangle$
2: $y_0 = -g$, $[X_0 = 0]$, $\langle S_0 = 0 \rangle$
3: **for** $k = 1, 2, 3, \dots$ **do**
4:     find $\eta_k \in \mathrm{argmin}_{\eta \geq 0} f(\eta y_k + (\eta - 1)g)$
5:     find $\lambda_k = \lambda_{\min}(\mathcal{G}^*(\nabla f(\eta_k y_k + (\eta_k - 1)g)))$ and associated normalized eigenvector $q_k$
6:     **if** $\lambda_k > -\epsilon$ **then**
7:         **break** with $y_\epsilon^\star = \eta_k y_k + (\eta_k - 1)g$, $[X_\epsilon^\star = \eta_k X_k]$, $\langle S_\epsilon^\star = \eta_k S_k \rangle$
8:     **else**
9:         choose step size $\theta_k > 0$
10:         $y_{k+1} = \eta_k y_k + (\eta_k - 1)g + \theta_k \mathcal{G}(q_k q_k^T)$
11:         $[X_{k+1} = \eta_k X_k + \theta_k q_k q_k^T]$, $\langle S_{k+1} = \eta_k S_k + \theta_k q_k (q_k^T \Omega) \rangle$
12:     **end if**
13:     (optional greedy step)
14: **end for**

---

The auxiliary variable $y = \mathcal{G}(X) - g$ drives the iterations. Because $X_0 = 0$, we know that $y_0 = -g$. It is simple to verify that finding $\eta_k$ in line 3 of Algorithm 1 is equivalent to minimizing $f(\eta y_k + (\eta - 1)g)$ over $\eta \geq 0$. So, as long as we have $y_k$, it is possible to compute $\eta_k$ without explicitly using $X_k$. It is also easy to verify that finding the direction of descent under the Schatten 1-norm (line 4 in Algorithm 1) is equivalent to finding the minimum eigenvalue and associated normalized eigenvector of $\mathcal{G}^*(\nabla f(\eta_k y_k + (\eta_k - 1)g))$. This can be carried out efficiently using the shifted power method or the Lanczos method because matrix-vector products can be efficiently evaluated (see [23]). Again, we do not need $\eta_k X_k$ explicitly to accomplish this step. If we use an exact or backtracking line search to choose $\theta_k$, it can be found by minimizing a simple function of $y_k$. Finally, the update step for finding $X_{k+1}$ is equivalent to $y_{k+1} = \mathcal{G}(\eta_k X_k + \theta_k q_k q_k^T) = \eta_k y_k + (\eta_k - 1)g + \theta_k \mathcal{G}(q_k q_k^T)$, which can be found efficiently because evaluating $\mathcal{G}$ on rank-1 matrices is easy.

Taken altogether, the previous discussion shows that the auxiliary variable $y_k$ can be used to run the iterations of conic descent without ever actually needing $X_k$ explicitly; it generates *exactly* the same sequence of points. This means that we are free to use memory-efficient data structures which only approximate $X_k$, and the error in these approximations will not affect the running of conic descent. For clarity, the expressions in Algorithm 2 with square brackets indicate the implicit iterates $X_k$. Following the lead of [21, 23, 24], we suggest using randomized matrix sketches; expressions with angled brackets indicate this option. Upon completion, we will have obtained a sketch $S_\epsilon^\star$ which can be used to obtain an approximation of $X_\epsilon^\star$. The details of this process and an error guarantee are in Appendix F. Roughly speaking, nearly perfect recovery of $X_\epsilon^\star$ is possible with high probability if its approximate rank is less than that of $r$.

Finally, we suggest using a greedy step based on the Burer-Monteiro factorization heuristic which greatly speeds up the convergence of CD in practice. Specifically, in line 13, we find a low-rank update using a point $(t_k, U_k)$ found from running a descent method (*e.g.*, conjugate gradient) on the problem

$$\underset{t \in \mathbf{R}, U \in \mathbf{R}^{n \times r}}{\text{minimize}} \quad f(\mathcal{G}(t^2 X_{k+1} + UU^T) - g) = f(t_k^2(y_{k+1} + g) + \mathcal{G}(UU^T) - g) \qquad (6)$$

We expand on this in Appendix G.

The memory requirement of Algorithm 2 with the use of sketches and the greedy step is $O(m + nr)$ because it is dominated by the storage for $y_k$, $S_k$, and $U_k$. This can be substantially lower than $O(n^2)$.

In the following two subsections, we examine the numerical performance of this method on two different problems: phase retrieval and positive semidefinite matrix completion. The use of the greedy step allows us to quickly reduce the objective value while the steps of conic descent ensure that we do not become trapped at suboptimal feasible points.

## 4.1  Examples and numerical results

**Phase retrieval**    In Appendix E, a simple example is contrived to demonstrate that CG can perform poorly relative to conic descent when a gross overestimate of $R$ is used. However, empirically, this can also occur when a relatively small overestimate of $R$ is used. We use phase retrieval as a demonstration because it is straightforward to obtain a reasonable estimate of $R$ using only the raw data.

Let $k$ be some small integer. Suppose that $x \in \mathbf{R}^n$ is a signal that we desire to reconstruct from the measurements $b_i = (a_i^T x)^2 + \epsilon_i, \quad i = 1, \ldots, kn$, where $\epsilon_1, \ldots, \epsilon_{kn}$ are independent and identically distributed Gaussian random variables with zero mean. The measurement vectors $a_i \in \mathbf{R}^n$ are the rows of the matrix $A = [DS_1 \quad \cdots \quad DS_k]^T$ where $D$ is the discrete cosine transform and $S_1, \ldots, S_k$ are diagonal matrices of independent random signs. Without any other assumptions, this problem is extremely difficult to solve in general, so we instead work with a convex relaxation. In this formulation, the quadratic measurements of the original signal are lifted to linear measurements on positive semidefinite matrices. The natural convex relaxation is given by

$$
\begin{aligned}
\text{minimize} \quad & \|\mathcal{A}(X) - b\|_2^2 + \gamma \mathbf{tr}(X) \\
\text{subject to} \quad & X \in \mathbf{S}_+^n
\end{aligned}
\tag{7}
$$

where $\mathcal{A}(X) = (a_1^T X a_1, \ldots, a_{kn}^T X a_{kn})$ and $\gamma$ is a tuning parameter used to promote low-rank solutions. After solving problem (7), the rank-1 approximation to the optimal solution is used as our estimate of the original signal.

We can form an estimate of $R$ by noting that

$$
\frac{1}{kn} \sum_{i=1}^{kn} b_i = \frac{1}{kn} \sum_{i=1}^{kn} x^T(a_i a_i^T)x + \frac{1}{kn} \sum_{i=1}^{kn} \epsilon_i \approx \frac{1}{kn} \sum_{i=1}^{kn} x^T(a_i a_i^T)x = \frac{1}{n}\|x\|_2^2.
$$

The approximation follows from the law of large numbers and because $A^T A = kI$. Thus, assuming that $X^\star$ is approximately rank-1, we expect that $\frac{1}{k}\mathbf{1}^T b \approx \mathbf{tr}(X^\star)$.

For our experiments, we used images in the CIFAR-10 dataset from [14] (after conversion to grayscale) as the raw signal and Gaussian noise with SNR of 20 dB. We used $k = 10$ and $\gamma = 5.0 \times 10^{-5}$. For CG, we used $R = \mathbf{1}^T b$ as an overestimate of the trace of $X^\star$. We used $r = 3$ for the sketch parameter. A line search was used to determine the step size for both CD and CG. The objective value and matrix multiplication count (for matrices of the form $\mathcal{A}^*(z)$ with $z \in \mathbf{R}^{kn}$) after 500 iterations of CG was recorded. The number of matrix multiplications required by CD to achieve the CG final objective value was also recorded. The histogram on the left in Figure 1 shows the CD to CG ratio of these matrix multiplication counts for 50 images. Clearly, CD requires at most roughly half as many to achieve a desired accuracy—and frequently a lot fewer. The use of the number of matrix-vector multiplications is a good metric because this is frequently the main computational cost in these problems. Finding the minimum eigenvalue using the Lanczos algorithm requires many calls to this method. In our experience, one reason for the good performance of CD is that the rescaling step creates a better conditioned minimum eigenvalue computation in the subproblem minimization step.

On the right in Figure 1, we show the reconstructions of a particular image achieved after halting the methods at approximately 1500 matrix-vector multiplications. There is a visible improvement in the quality of the reconstruction obtained from CD over that obtained from CG. Additionally, the bottom right image shows the reconstruction obtained from conic descent with a greedy step every 100 iterations. In all three instances, the same sketch parameter $\Omega$ was used so that the differences in the images are the result of the differences in the iterate itself and not in the sketch.

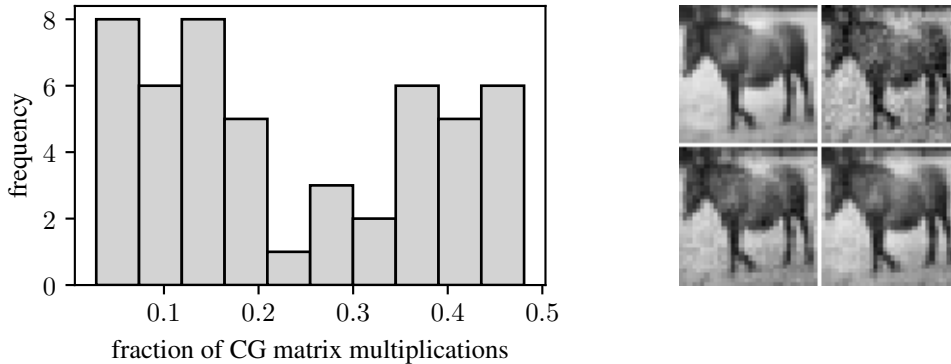

Figure 1: Left: Fraction of the number of matrix-vector multiplications required by CD to find a point which has the same objective value as found by CG after 500 iterations for 50 different grayscale images from the CIFAR-10 dataset. Right: Image of a horse from the CIFAR-10 dataset. Clockwise from upper left: The original image, CG reconstruction, CD with a greedy step reconstruction, and CD reconstruction after using approximately 1500 matrix-vector multiplications.

We note that it is probably more reasonable to use something like $R = \frac{2}{k}\mathbf{1}^T b$ as an overestimate. If we use such a bound, numerical results indicate that CG actually outperforms CD (without the greedy heuristic) in terms of iterations required to hit a desired objective value. The point of the above experiments is to demonstrate the risks of using overestimates of the optimal bound with CG. For this particular phase retrieval problem it is relatively easy to obtain a good estimate for this bound, but in general it is more difficult, and even getting it within a factor of ten can be a challenge for different objectives. In the next section, we focus on the comparison of CD with and without the greedy heuristic.

**Positive semidefinite matrix completion**    In this section, we examine the numerical performance of CD with the greedy heuristic as we vary the rank $r$ of its updates. We will construct an example for which the Burer-Monteiro factorization heuristic (with a good initialization) by itself performs poorly, but as part of the greedy step in CD, performs significantly better.

Suppose that we are given a random sampling of noisy measurements of entries from some low-rank positive semidefinite matrix $A$ and that we would like to construct an approximation to $A$ using these measurements. More specifically, suppose that for some index set $\mathcal{I}$, we obtain the measurements $b_{ij} = A_{ij} + \epsilon_{ij}$ for $(i, j) \in \mathcal{I}$ where $\epsilon_{ij}$ are independent and identically distributed Gaussian random variables with zero mean. This problem arises (in varying forms) in many different applications and usually goes by the name of matrix completion or matrix sensing. We want to solve the problem

$$\begin{array}{ll} \text{minimize} & \sum_{(i,j)\in\mathcal{I}}(X_{ij} - b_{ij})^2 \\ \text{subject to} & X \in \mathbf{S}_+^n. \end{array} \tag{8}$$

This problem is directly amenable to our memory-efficient version of CD. For our purposes, we suppose that $A = VV^T$ for some $V \in \mathbf{R}^{100\times 3}$. Unlike many other matrix sensing problems, we do not sample uniformly over $i$ and $j$. Instead, we sample every entry in the upper left $10 \times 10$ block and with probability 0.1 for each entry in the remainder. This is a toy model of correlated sampling and is more challenging than the standard uniform sampling. Because of the addition of Gaussian noise, this means that we are directly sampling a matrix with rank approximately 13. For our experiments, we used Gaussian noise with SNR of 20 dB. The greedy step was taken for the first iteration and every hundred iterations thereafter. All greedy step subproblems were solved using the conjugate gradient method and terminated once the 2-norm of the gradient in problem (6) was less than $10^{-6}$. Problem (8) was also solved using the Splitting Conic Solver (see [18]) that ships with CVXPY (see [1, 8]) to obtain the optimal value of the problem $p^\star$ as a reference against which to compare.

The results for a typical instance with varying $r$ are displayed together on the left side of Figure 2. The dots mark the objective value achieved after a single step of CD followed by the greedy heuristic; this corresponds to initializing a BM factorization heuristic method with the result of one step of CD

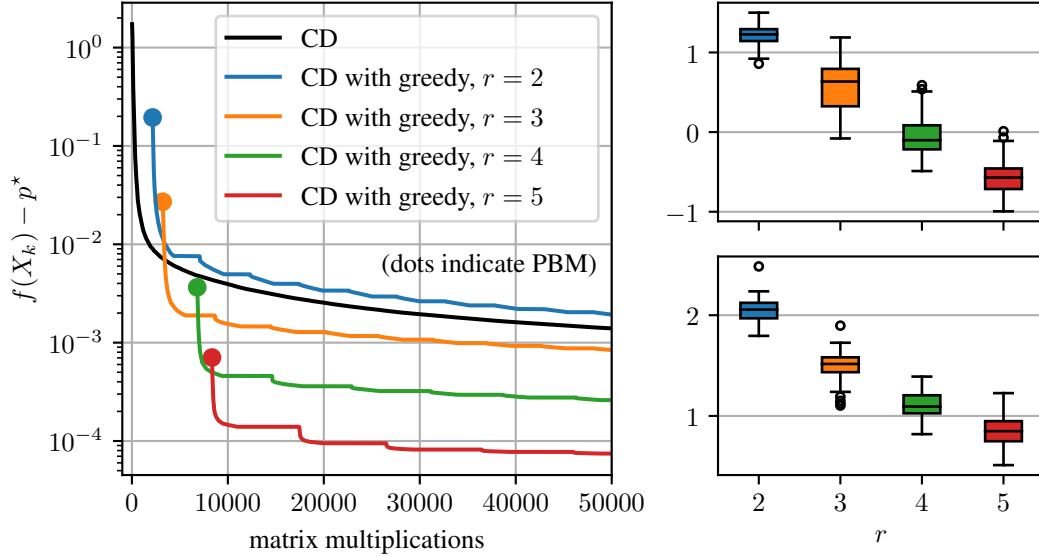

Figure 2: Left: The convergence of the greedy heuristic for CD applied to an instance of problem (8) for varying rank $r$ of the greedy step. The solid dot on the colored lines indicates the objective value obtained by a single greedy step after a single step of CD. The greedy step is taken after the first iteration of CD and every hundred iterations thereafter. Top right: The box plots generated from 50 random instances of our matrix completion problem of the difference in the base-10 logarithms of the optimal value obtained from PBM and the value of CD (without the greedy heuristic) after approximately the same number of matrix multiplications. This corresponds to the vertical distance on the left between the colored dots and the black line; note the logarithmic scale. Negative numbers indicate that the PBM solution was more accurate. Bottom right: The difference in the base-10 logarithms of the optimal value obtained from PBM and the optimal value obtained by CD with a greedy heuristic of the same rank after 50,000 matrix multiplications. This corresponds to the vertical distance on the left between the colored dot and the value at 50,000 matrix multiplications of the line of the same color.

and will serve as our reference performance for a pure BM-based method (herafter denoted PBM). For $r = 2$ or $r = 3$, we see that CD without the greedy step obtains a smaller objective value than PBM for the same number of matrix multiplications. For $r = 4$ or $r = 5$, PBM does at least as well as CD for the same number of matrix multiplications but at the cost of using more memory. We also see that, in all cases, the use of the greedy step in CD significantly improves on the performance of PBM because CD (with or without the greedy heuristic) is *guaranteed* to converge to the optimal objective value. Thus, the results suggest that in situations which are highly memory-constrained, the use of CD with the greedy heuristic can provide significantly better solutions than PBM while using about the same memory. Taken together, it appears that CD with greedy steps provides a way to fuse the two popular methods of dealing with memory-limited optimization over the positive semidefinite cone.

On the top right side of Figure 2, we show box plots of the difference between the objective values obtained by CD without the greedy step and PBM (*i.e.*, the vertical distance between the colored dot and the black line on the left side). On the bottom right side of Figure 2, we show box plots of the difference between the final objective value obtained by CD with the greedy step and PBM (*i.e.*, the vertical distance between the colored dot and the value obtained at 50,000 matrix multiplications. The box plots were created from 50 instances of problem (8). The vertical distances measured in these box plots correspond roughly to the improvement in the number of digits of accuracy. These show that the performance on the left in Figure 2 is typical of instances of this problem.

## Broader Impact

Because CD is guaranteed to converge to the globally optimal solution of problem (1), it can be used in safety critical applications of machine learning that require guarantees of reliability.

## Acknowledgments and Disclosure of Funding

Andrew Naber gratefully acknowledges support from the Stanford Graduate Fellowship and the G.I. Bill (U.S. Department of Veterans Affairs). Oliver Hinder acknowledges support from the Dantzig-Lieberman Operations Research Fellowship.

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
