[Supplementary Material]

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

# A  Proof of Theorem 1

First, an upper bound on the objective value after one iteration of the method is obtained:

$$f(x_{k+1}) \leq f(\eta_k x_k) + \nabla f(\eta_k x_k)^T (x_{k+1} - \eta_k x_k) + \frac{L}{2} \|x_{k+1} - \eta_k x_k\|^2$$

$$= f(\eta_k x_k) + \theta_k \nabla f(\eta_k x_k)^T v_k + \frac{L}{2} \theta_k^2$$

$$\leq f(\eta_k x_k) + \theta_k \nabla f(\eta_k x_k)^T \frac{x^\star}{\|x^\star\|} + \frac{L}{2} \theta_k^2$$

$$\leq f(\eta_k x_k) + \frac{\theta_k}{\|x^\star\|} (f(x^\star) - f(\eta_k x_k)) + \frac{L}{2} \theta_k^2.$$

The first inequality follows from the upper bound implied by the assumption of a Lipschitz continuous gradient of $f$. The second inequality follows from the definition of $v_k$ as the minimizer of the subproblem in iteration $k$. The final inequality follows from the fact that $\nabla f(\eta_k x_k)^T (\eta_k x_k) = 0$ and the convexity of the objective function. Because $f(\eta_k x_k) \leq f(x_k)$, a simple rearrangement gives that

$$f(x_{k+1}) - f(x^\star) \leq \left(1 - \frac{\theta_k}{\|x^\star\|}\right) (f(x_k) - f(x^\star)) + \frac{L}{2} \theta_k^2.$$

For the sake of argument, suppose that $\theta_k = \frac{2\|x^\star\|}{k+2}$; in using a line search to determine $\theta_k$, the objective value will decrease by a larger amount at each iteration, and all our arguments still hold. Letting $\epsilon_k = f(x_k) - f(x^\star)$, we have that

$$\epsilon_{k+1} \leq \left(1 - \frac{2}{k+2}\right) \epsilon_k + \frac{L}{2} \left(\frac{2\|x^\star\|}{k+2}\right)^2.$$

We now present an inductive argument to show the result: $\epsilon_k \leq \frac{2L\|x^\star\|^2}{k+2}$ for all $k \geq 1$. For $k = 1$,

$$\epsilon_1 \leq \frac{L\|x^\star\|^2}{2} \leq \frac{2L\|x^\star\|^2}{3}.$$

Now suppose that the result holds for some $k \geq 1$. Then,

$$\epsilon_{k+1} \leq \left(1 - \frac{2}{k+2}\right) \frac{2L\|x^\star\|^2}{k+2} + \frac{L}{2} \left(\frac{2\|x^\star\|}{k+2}\right)^2$$

$$= \frac{2L\|x^\star\|^2(k+1)}{(k+2)^2}$$

$$\leq \frac{2L\|x^\star\|^2}{k+3}.$$

The final inequality follows because $(k+2)^2 \geq (k+1)(k+3)$.

# B  Proof of Theorem 2

For clarity, we define our backtracking line search in Algorithm 3. Recall that $\nabla f(\eta_k x_k)^T v_k$ is the optimal value of the subproblem computed in step 4 of Algorithm 1 and so is readily accessible at each iteartion. Also, recall that $\nabla f(\eta_k x_k)^T v_k < 0$; otherwise, CD will have already terminated.

---

**Algorithm 3** Backtracking line search to determine $\theta_k$

---

1: $\theta_k = -\nabla f(\eta_k x_k)^T v_k$
2: **while** $f(\eta_k x_k + \theta_k v_k) > f(\eta_k x_k) + \alpha \theta_k \nabla f(\eta_k x_k)^T v_k$ **do**
3:     $\theta_k \leftarrow \beta \theta_k$
4: **end while**

---

There are two cases to consider. In the first case, the backtracking line search terminates immediately with $\theta_k = \nabla f(\eta_k x_k)^T v_k$, so

$$f(x_{k+1}) \leq f(\eta_k x_k) - \alpha(\nabla f(\eta_k x_k)^T v_k)^2. \tag{9}$$

In the second case, the stopping criterion is not initially satisfied. We can use the quadratic upper bound implied by Lipschitz continuity of the gradient to place a lower bound on $\theta_k$ in this case. Because

$$f(\eta_k x_k + \theta v_k) \leq f(\eta_k x_k) + \theta \nabla f(\eta_k x_k)^T v_k + \frac{L}{2}\theta^2 \leq f(\eta_k x_k) + \alpha \theta \nabla f(\eta_k x_k)^T v_k$$

for all $\theta$ such that

$$0 \leq \theta \leq -\frac{2(1-\alpha)}{L}\nabla f(\eta_k x_k)^T v_k,$$

we know that $\theta_k \geq -\frac{2(1-\alpha)\beta}{L}\nabla f(\eta_k x_k)^T v_k$. Otherwise, the stopping criterion will have been triggered; this must occur after at most $\frac{\log(2(1-\alpha)/L)}{\log \beta}$ iterations of the while loop. Thus,

$$\begin{aligned} f(x_{k+1}) &\leq f(\eta_k x_k) + \alpha \theta_k \nabla f(\eta_k x_k)^T v_k \\ &\leq f(\eta_k x_k) - \frac{2\alpha(1-\alpha)\beta}{L}(\nabla f(\eta_k x_k)^T v_k)^2. \end{aligned} \tag{10}$$

We combine (9) and (10) to obtain

$$\begin{aligned} f(x_{k+1}) &\leq f(\eta_k x_k) - \min\{\alpha, 2\alpha(1-\alpha)\beta/L\}(\nabla f(\eta_k x_k)^T v_k)^2 \\ &= f(\eta_k x_k) - \min\{2L\alpha, 4\alpha(1-\alpha)\beta\}\frac{1}{2L}(\nabla f(\eta_k x_k)^T v_k)^2 \\ &= f(\eta_k x_k) + \min\{2L\alpha, 4\alpha(1-\alpha)\beta\}\inf_{\theta \geq 0}\left(\theta \nabla f(\eta_k x_k)^T v_k + \frac{L}{2}\theta^2\right). \end{aligned}$$

Let $C = \min\{2L\alpha, 4\alpha(1-\alpha)\beta\}$. So, for any $\theta \geq 0$, we have that

$$\begin{aligned} f(x_{k+1}) &\leq f(\eta_k x_k) + C\left(\theta \nabla f(\eta_k x_k)^T v_k + \frac{L}{2}\theta^2\right) \\ &\leq f(\eta_k x_k) + C\left(\theta \nabla f(\eta_k x_k)^T \frac{x^\star}{\|x^\star\|} + \frac{L}{2}\theta^2\right) \\ &\leq f(\eta_k x_k) + C\left(\frac{\theta}{\|x^\star\|}(f(x^\star) - f(\eta_k x_k)) + \frac{L}{2}\theta^2\right). \end{aligned}$$

The second inequality follows from the definition of $v_k$ as the minimizer of the subproblem in iteration $k$. The final inequality follows from the fact that $\nabla f(\eta_k x_k)^T(\eta_k x_k) = 0$ and the convexity of the objective function. Because $f(\eta_k x_k) \leq f(x_k)$, a simple rearrangement gives that

$$f(x_{k+1}) - f(x^\star) \leq \left(1 - C\frac{\theta}{\|x^\star\|}\right)(f(x_k) - f(x^\star)) + C\frac{L}{2}\theta^2.$$

For the sake of argument, suppose that $\theta = \frac{2\|x^\star\|}{C(k+2)}$. Letting $\epsilon_k = f(x_k) - f(x^\star)$, we have that

$$\epsilon_{k+1} \leq \left(1 - \frac{2}{k+2}\right)\epsilon_k + \frac{L}{2C}\left(\frac{2\|x^\star\|}{k+2}\right)^2.$$

An inductive argument nearly identical to that in the proof of Theorem 1 gives the result:

$$\epsilon_k \leq \frac{2L\|x^\star\|^2}{C(k+2)}$$

for all $k \geq 1$.

## C Proof of Theorem 3

First, an upper bound on the objective value after one iteration of the method is obtained:

$$f(x_{k+1}) \leq f(\eta_k x_k) + \nabla f(\eta_k x_k)^T (x_{k+1} - \eta_k x_k) + \frac{L}{2} \|x_{k+1} - \eta_k x_k\|^2$$

$$= f(\eta_k x_k) + \theta_k \nabla f(\eta_k x_k)^T \tilde{v}_k + \frac{L}{2} \theta_k^2$$

$$\leq f(\eta_k x_k) + \theta_k \nabla f(\eta_k x_k)^T v_k + \theta_k \frac{\delta}{k+2} + \frac{L}{2} \theta_k^2$$

$$\leq f(\eta_k x_k) + \theta_k \nabla f(\eta_k x_k)^T \frac{x^\star}{\|x^\star\|} + \theta_k \frac{\delta}{k+2} + \frac{L}{2} \theta_k^2$$

$$\leq f(\eta_k x_k) + \frac{\theta_k}{\|x^\star\|} (f(x^\star) - f(\eta_k x_k) + \nabla f(\eta_k x_k)^T (\eta_k x_k)) + \theta_k \frac{\delta}{k+2} + \frac{L}{2} \theta_k^2$$

$$\leq f(\eta_k x_k) + \frac{\theta_k}{\|x^\star\|} (f(x^\star) - f(\eta_k x_k)) + \frac{\theta_k}{\|x^\star\|} \left( \frac{\gamma}{k+2} \right) + \theta_k \frac{\delta}{k+2} + \frac{L}{2} \theta_k^2.$$

The first inequality follows from the upper bound implied by the assumption of a Lipschitz continuous gradient of $f$. The second inequality follows from the definition of $\tilde{v}_k$ as the approximate minimizer of the subproblem (*i.e.*, inequality (4)). The third inequality follows from the fact that $v_k$ is the optimal solution of the subproblem. The fourth inequality follows from the convexity of $f$. The final inequality follows from condition (3). Because $f(\eta_k x_k) \leq f(x_k)$, a simple rearrangement gives that

$$f(x_{k+1}) - f(x^\star) \leq \left( 1 - \frac{\theta_k}{\|x^\star\|} \right) (f(x_k) - f(x^\star)) + \left( \frac{\gamma}{\|x^\star\|} + \delta \right) \frac{\theta_k}{k+2} + \frac{L}{2} \theta_k^2.$$

For the sake of argument, suppose that $\theta_k = \frac{2\|x^\star\|}{k+2}$; in using a line search to determine $\theta_k$, the objective value will decrease by a larger amount at each iteration, and all our arguments still hold. Letting $\epsilon_k = f(x_k) - f(x^\star)$, we have that

$$\epsilon_{k+1} \leq \left( 1 - \frac{2}{k+2} \right) \epsilon_k + \frac{1}{2} \left( \frac{\gamma}{\|x^\star\|^2} + \frac{\delta}{\|x^\star\|} + L \right) \left( \frac{2\|x^\star\|}{k+2} \right)^2.$$

An inductive argument nearly identical to that in the proof of Theorem 1 gives the result:

$$\epsilon_k \leq \frac{2}{k+2} (\gamma + \delta\|x^\star\| + L\|x^\star\|^2)$$

for all $k \geq 1$.

## D Proof of Theorem 4

At each iteration, $\eta_k x_k \in K$ and $\nabla f(\eta_k x_k)^T (\eta_k x_k) = 0$, so it remains to bound the iterations required to obtain $\mathbf{dist}^* (\nabla f(\eta_k x_k), K^*) \leq \epsilon$ (equivalently, $\nabla f(\eta_k x_k)^T v_k \geq -\epsilon$). As in the proof of of Theorem 1, the quadratic upper bound implied by Lipschitz continuity of the gradient gives that

$$f(x_{k+1}) \leq f(\eta_k x_k) + \theta_k \nabla f(\eta_k x_k)^T v_k + \frac{L}{2} \theta_k^2$$

$$= f(\eta_k x_k) - \frac{1}{2L} (\nabla f(\eta_k x_k)^T v_k)^2$$

$$\leq f(x_k) - \frac{1}{2L} (\nabla f(\eta_k x_k)^T v_k)^2.$$

The equality follows from the stepsize choice $\theta_k = -\frac{1}{L} \nabla f(\eta_k x_k)^T v_k$, and the final inequality follows because $f(\eta_k x_k) \leq f(x_k)$. So, at each iteration, the function value decreases. Of course, a line search will lead to larger decreases in the objective function and the result will continue to hold. By collapsing the differences in function values after $N$ iterations,

$$f(x_0) - f(x_N) \geq \frac{1}{2L} \sum_{k=1}^{N} (\nabla f(\eta_k x_k)^T v_k)^2.$$

Figure 3: Solving problem (11) using the conditional gradient method (with the additional feasible constraint $\|x\|_1 \le 10^8$) and conic descent for $n = 25$.

Finally, $f(x_0) - f(x^\star) \ge f(x_0) - f(x_N)$, so the summation is bounded above for every $N$, and we can conclude that $\lim_{k \to \infty} (\nabla f(\eta_k x_k)^T v_k)^2 = 0$. Recall that at each iteration of the algorithm, $\nabla f(\eta_k x_k)^T v_k < 0$ (or else it will have already terminated). Furthermore, this has established that

$$\min_{k=1,\dots,N} (\nabla f(\eta_k x_k)^T v_k)^2 \le \frac{2L}{N}(f(x_0) - f(x^\star)),$$

and the result follows.

## E  Simple example for which CG stalls

We wil show that for the problem

$$\begin{aligned} \text{minimize} \quad & \tfrac{1}{2}x^T(I + \mathbf{1}\mathbf{1}^T)x - \mathbf{1}^T x \\ \text{subject to} \quad & x \in \mathbf{R}^n_+, \end{aligned} \tag{11}$$

the CG method with an overestimate of $R$ stalls.

Since the gradient of the objective is given by $(I + \mathbf{1}\mathbf{1}^T)x - \mathbf{1}$, we can verify by substitution that $x^\star = \frac{1}{n+1}\mathbf{1}$ and $p^\star = -\frac{n}{2(n+1)}$. To use the conditional gradient method to solve this problem, we must add the feasible constraint $\|x\|_1 \le R$ where $R \ge \frac{n}{n+1}$. Suppose that we use a gross overestimate of $R$, *i.e.*, we effectively let $R \to \infty$. In this case, we will sketch an argument that the conditional gradient method stalls. Specifically, for $k = 1, \dots, n$, the iterate $x_k \approx \sum_{i=1}^{k} 2^{-i}e_i$ which implies that $f(x_k) \approx \frac{1}{3}(4^{-k} - 1) > -\frac{1}{3}$ even though $p^\star \approx -\frac{1}{2}$. See Figure 3 for a small numerical example.

At each iteration, the subproblem minimization in CG returns with either $v_k = 0$ if all entries of $\nabla f(x_k)$ are nonnegative or $v_k = Re_i$ where $i$ is the index of the first occurrence of the minimum entry of $\nabla f(x_k)$ otherwise. For the step size determination we use a line search.

To show that $x_k \approx \sum_{i=1}^{k} 2^{-i}e_i$ for $k = 1, \dots, n$, we proceed inductively. At the initial point $x_0 = 0$, the gradient $\nabla f(x_0) = -\mathbf{1}$ and the subproblem minimization returns with $v_0 = Re_1$. The line search

returns with $\theta_0 = \frac{1}{2R}$, so $x_1 = 2^{-1}e_1$. Now, suppose that $x_k \approx \sum_{i=1}^{k} 2^{-i}e_i$ for some $k \leq n-1$. We have that

$$\nabla f(x_k) = x_k + (\mathbf{1}^T x_k - 1)\mathbf{1}$$
$$\approx x_k - 2^{-k}\mathbf{1},$$

using the well-known formula for the sum of a finite geometric sequence. The subproblem minimization returns with $v_k = Re_{k+1}$. The line search returns with $\theta_k \approx \frac{2^{-(k+1)}}{R}$. To see this, set the derivative of $f(x_k + \theta(v_k - x_k))$ with respect to $\theta$ to zero and solve:

$$\theta_k = \frac{(\mathbf{1}^T x_k - 1)\mathbf{1}^T (v_k - x_k) - (v_k - x_k)^T x_k}{\|v_k - x_k\|_2^2 + (\mathbf{1}^T(v_k - x_k))^2}$$
$$= \frac{(\mathbf{1}^T x_k - 1)\mathbf{1}^T (v_k - x_k) + \|x_k\|_2^2}{\|v_k - x_k\|_2^2 + (\mathbf{1}^T(v_k - x_k))^2}$$
$$\approx \frac{2^{-k}\mathbf{1}^T (v_k - x_k) + \|x_k\|_2^2}{\|v_k - x_k\|_2^2 + (\mathbf{1}^T(v_k - x_k))^2}$$
$$\approx \frac{2^{-k}R}{R^2 + R^2}$$
$$= \frac{2^{-(k+1)}}{R}.$$

The first approximation follows because $\mathbf{1}^T x_k \approx 1 - 2^{-k}$ (again using the formula for the sum of a finite geometric sequence). The second approximation follows because $R$ is (by assumption) much larger than any other parameter in this problem. Thus, $x_{k+1} = (1 - \theta_k)x_k + \theta_k v_k \approx \sum_{i=1}^{k+1} 2^{-i}e_i$.

Figure 3 shows the convergence of both CD and CG (with $R = 10^8$) for $n = 25$. Both methods used line searches and similar subproblem routines. Clearly, as predicted, CG stalls for the first $n$ iterations. Interestingly, it appears to repeatedly stall during the course of the next $2n$ iterations as well.

Why doesn't CD stall? The re-scaling at each iteration has the effect of taking a step toward the origin which CG does not do until after the first $n$ iterations — even though the first iterate is significantly further from the origin than the optimal solution. The purpose of this example is to demonstrate that CD is not simply recovered by using CG with a large bound and that using such a bound can be detrimental.

Finally, we note that the magnitude of $R$ required to exhibit this behavior grows rapidly with $n$. This can cause numerical issues, and the step size approximations above will no longer be valid in finite precision arithmetic. However, the staircase behavior (where CG stalls and fails to take a step toward the origin for a large number of iterations) can also appear in practical problems with reasonable overestimates of $R$. In Figure 4, the convergence of both conditional gradient and conic descent are shown for the first 100 iterations of an instance of our phase retrieval problem for the CIFAR-10 dataset (as described in subsection 4.1). As we saw in the simple example, CG appears to stall repeatedly, though not quite as extremely.

# F   Sketch reconstruction

Recall that the expressions in Algorithm 2 with angled brackets indicate the steps necessary to track the iterates using a sketch in a memory-efficient manner. To use a randomized sketch, we first fix a matrix $\Omega \in \mathbf{R}^{n \times r}$ such that each entry is drawn independently from a standard normal distribution. The parameter $r$ is assumed to be much smaller than $n$ and controls the error in our approximation. The sketch of the iterate $X_k$ is $S_k = X_k \Omega$. The linearity of the sketch (as a function of $X_k$) allows us to efficiently form the sketch of $X_{k+1}$ in line 11.

Upon termination, we have a sketch $S_\epsilon^\star$ of $X_\epsilon^\star$, and its approximation is given by

$$\hat{X}_\epsilon^\star = (X_\epsilon^\star \Omega)(\Omega^T X_\epsilon^\star \Omega)^\dagger (X_\epsilon^\star \Omega)^T$$
$$= S_\epsilon^\star (\Omega^T S_\epsilon^\star)^\dagger S_\epsilon^{\star T}.$$

Figure 4: Iterate objective value for the first 100 iterations of conic descent versus conditional gradient on the phase retrieval problem for the image in Figure 1

The (thin) eigenvalue decomposition of $\hat{X}_\epsilon^\star = Q\Lambda Q^T$ with $Q \in \mathbf{R}^{n \times r}$ and $\Lambda \in \mathbf{R}^{r \times r}$ can be found via Algorithm 4. The average error in this approximation is bounded as

$$\mathbf{E} \, \|X_\epsilon^\star - \hat{X}_\epsilon^\star\|_{S_1} \leq \left(1 + \frac{R}{r - R - 1}\right) \sum_{i=R+1}^{n} \sigma_i(X_\epsilon^\star),$$

and a similar bound holds with high probability; see [10, 21] or Appendix B of [23] for more detailed discussions of the probabilistic guarantees of sketching positive semidefinite matrices. This means that, for example, if $X_\epsilon^\star$ is truly rank-$r$, then the error in the approximation is very small with high probability.

---

**Algorithm 4** Sketch reconstruction

---

1: find Cholesky decomposition $LL^T$ of $\Omega^T S_\epsilon^\star$
2: find the (thin) singular value decomposition $U\Sigma V^T$ of $S_\epsilon^\star L^\dagger$
3: $Q = U, \Lambda = \Sigma^2$

---

## G   Greedy step details

Algorithm 5 details our proposed greedy step. Note that the nonnegativity of $t_k^2$ and the positive semidefiniteness of $U_k U_k^T$ guarantee that the updated $X_{k+1}$ remains feasible. The greedy step does not need to be taken at every iteration of Algorithm 2; it can just as well be called at some regular interval (*e.g.*, every hundredth iteration). For the first iteration, a random initialization for the descent method is used. Subsequent initializations modify the previous point found from the descent method in accordance with line 4. There are other reasonable initializaiton schemes, but we have found this one to work well in practice. The rank of the update $r$ is assumed to be much less than $n$.

---
**Algorithm 5** Greedy step (line 13 in Algorithm 2)
---
1: **if** $k = 0$ **then**
2:     form $\tilde{U} \in \mathbf{R}^{n \times r}$ with entries independently drawn from a standard normal distribution
3: **else**
4:     replace the column of $\tilde{U}$ that has smallest norm with $q_k$
5: **end if**
6: find $(t_k, U_k)$ by running a descent method initialized at $(1, \tilde{U})$ on problem (6)
7: $y_{k+1} \leftarrow t_k^2(y_{k+1} + g) + \mathcal{G}(U_k U_k^T) - g$
8: $[X_{k+1} \leftarrow t_k^2 X_{k+1} + U_k U_k^T], \langle S_{k+1} \leftarrow t_k^2 S_{k+1} + U_k(U_k^T \Omega) \rangle$
9: $\tilde{U} \leftarrow U_k$
---