[Reviews · NeurIPS 2020]

Review 1

Summary and Contributions: This is a very nice extension of recent work on matrix-sketching in conic optimisation (https://arxiv.org/abs/1912.02949). The authors simplify the original approach (https://arxiv.org/abs/1912.02949), which considered randomised projections within conditional gradient (https://arxiv.org/abs/1302.2325, http://proceedings.mlr.press/v97/yurtsever19a.html), a variant of the projected gradient. The simplified algorithm, called "conic descent", is genuinely elegant. They prove the O(1/k) convergence rate, under rather strong assumptions, which are satisfied e.g. in matrix completion. The memory requirements seem, indeed, low, but the computational results are not very convincing.

Strengths: The authors present a very elegant algorithm and prove its convergence rate in a practically relevant special case. The code provided in the supplementary material is quite elegant, although the computational results in the main body of the paper seem rather muddled.

Weaknesses: The key claims related to the matrix sketching are NOT proven, or even stated formally. In the main body of the text, they are hinted at (line 184), with a reference to Appendix F, but there, there is a statement "The average error in this approximation is bounded as" with an expression "out of the blue", with no derivation or proof or reference. The algorithm of Section 5 considers a very special case of semidefinite programming, but does not relate that to other special cases studied in the literature. Specifically, the authors say (l. 44) that Burer-Monteiro is applicable under "various strong assumptions on f", but some of these "strong assumptions seem less strong than what the authors utilise. Given the above, it seems somewhat riling that the authors keep talking about "Burer Monteiro heuristic" -- when they do not talk about the "Conic Descent heuristic". ;-) For the very general SDPs, both are, indeed, heuristics. The empirical results are very basic. Considering that the authors have a wealth of beautiful material in the appendix, e.g., Appendix F on the Sketch reconstruction, it may be worth replacing the half-backed empirical results with some good theory.

Correctness: Correct as far as I can tell.

Clarity: Some parts (e.g. lines 92 - 102) are supercondensed and not clear even to an SDP specialist, the less being readily readable for the NIPS audience. Some of the language could be improved ("using the same order of memory", l. 279).

Relation to Prior Work: Generally, the authors have many relevant references in their bibtex, but the discussion is lacking. The authors should comment in more detail on the Nystrom sketches, and their origin. The authors should comment in more detail on the relationship of Orthogonal Matching Pursuit and the work of Jaggi et al [16] and the "conditional gradient on the Augmented Lagrangian". The authors should explain the relationship of conditional gradient and their conic descent to bundle methods and the conic bundle: https://www-user.tu-chemnitz.de/~helmberg/ConicBundle/ in particular. The authors discuss the relationship to the CGAL on lines 48-57, mention that the "trace constraint on the feasible set is limiting", but then introduce the objective in Section 4, which is closely related. The discussion of lines 48-57 would benefit from an explanation of the cases ok to study with the formulation (5) but not with the very strong assumptions of [23]. Some of the papers (http://proceedings.mlr.press/v54/yurtsever17a.html) are cited as arxiv pre-prints, while they have appeared already.

Reproducibility: Yes

Additional Feedback: I have read the other reviews and the rebuttal. I would prefer not to change my score, but I would tend to change it downwards, should there be a change. I like the direction, but not (so much) the paper.


Review 2

Summary and Contributions: In the draft, the author proposed conic descent, a conditional gradient method with a conic scaling step. They showed O(1/k) convergence rate to optimum without dependence on the geometry of the cone. They also provide memory efficient modification / sketching variants of the algorithm and showed in the experiment that the proposed algorithm is better than the conditional gradient method.

Strengths: The analysis of conic descent is elegant, independent of the shape of the cone, and it naturally applies to the memory-efficient version. I think this is enough for acceptance.

Weaknesses: The experiments are rather weak. The algorithm compares only to itself in the main paper and only to the conditional gradient method in the appendix. Further, the authors only showed the number of iteration vs. the optimality gap in the comparison. The catch is that the conic descent method performs one more exact search step than the conditional gradient method. Thus, comparing only the iteration counts is not fair. Judging from the current experiments, I believe that the practical acceleration of the method to CG is not that obvious, since the proposed method doesn't provide much acceleration in the phase retrieval task (when measuring iteration counts). Further, the proposed memory-efficient algorithm was not really memory-efficient in your experiments, for a reason similar to BFGS. The proposed algorithms required much more than 1k iterations to achieve an acceptable suboptimality, which means that it needed to store at least 1k vectors. The memory requirement for the 1k vectors exceeded the size of the full dense matrix (for the 1024 pixels of CIFAR-10 images), so it is not really memory-efficient. The current experiments didn't show the benefit of exploiting the low-rank structure in the solution. In this case, a comparison to traditional dense SDP solvers will be preferable.

Correctness: I believe the theory is correct.

Clarity: The paper is well-written.

Relation to Prior Work: The prior works are properly discussed.

Reproducibility: Yes

Additional Feedback: I have read the rebuttal and decided to keep my current score. For comparison to CG, I mean experiments in Figure 4 in the appendix. After accounting the 2x matrix-vector operations, the CG would outperform CD. The author's feedback for Figure 1 corresponds to the performance of CG when the radius is over-estimated (stated in line 239), not the overall performance of CG. For the author's feedback on memory efficiency, I agree that the version with a sketch is memory efficient (with limited accuracy), but the version without a sketch is indeed not memory efficient according to the author's use case.


Review 3

Summary and Contributions: This paper presents an interesting new algorithm that solves a niggling issue in many applications of the conditional gradient method when the domain is not naturally compact and no bound on the norm of a solution is known. I'm not aware of another approach with guaranteed convergence for this class of problems that does not require a bound on the norm of the solution.

Strengths: The paper solves an interesting and important problem in large scale semidefinite programming. The methods are clearly described, interesting, and the experiments and theory seem sound if rather standard.

Weaknesses: The greedy steps using the BM heuristic provide a nice speedup in practice, but from a theoretical standpoint it's a rather cheap trick. Still, I guess it's good to know that the provable and not provable steps may be interleaved without ill effect. However, the numerics on this question leave something to be desired. The left plot of figure 2 shows that the method is slow without the nonconvex BM steps. The convex outer method here acts as a safeguard on the BM method to guarantee convergence. But did the authors try simply running the BM method without the safeguard? My guess is that the method is likely to converge to the optimum even without the outer safeguard. Can you provide any numerical evidence that this new method can solve problems that tradition BM (eg with gradient descent) cannot solve? The assumption of no nonzero direction of recession is rather obtuse. I wonder if you could simply replace it by the assumption that the solution x^* is attained? (I suspect you cannot.) It seems like an unfortunate restriction. Are there important problems that it excludes?

Correctness: Yes, seems correct to me.

Clarity: Yes, easy to read.

Relation to Prior Work: Yes, relevant work is discussed. One useful addition might be to compare the nonconvex step to https://arxiv.org/abs/0807.4423.

Reproducibility: Yes

Additional Feedback: * line 172: you might cite [22] for details on Lanczos and the shifted power method in this setting. * line 187 nonconvex heuristic: this idea has a long history. Can you compare eg to the approach in https://arxiv.org/abs/0807.4423? * eqn (7) has a penalty to ensure f has no nonzero direction of recession. Yet eqn (8) does not. Can you say why the penalty is not needed to ensure no nonzero direction of recession in eqn (8)? * The left plot of figure 2 shows that the method is slow without the nonconvex BM steps. The convex outer method here acts as a safeguard on the BM method to guarantee convergence. But did the authors try simply running the BM method without the safeguard? My guess is that the method is likely to converge to the optimum even without the outer safeguard. Can you provide any numerical evidence that this new method can solve problems that tradition BM (eg with gradient descent) cannot solve?


Review 4

Summary and Contributions: In this paper, the authors consider conditional gradient methods for optimization problems where the constraint is a cone. They introduce a few variants that incorporates line search into the conic descent method. They apply the method to solving problems with PSD cone constraints, and show that the proposed approaches are memory efficient. From a practical point of view, they use the Burer-Monteiro heuristic to speed up convergence. They apply the method to phase retrieval and PSD matrix completion and show that it outperforms conditional gradient.

Strengths: The paper is well written, and is relevant to the community. The methods proposed are well motivated, and empirical results suggest they are better in both speed and performance when compared to conditional gradient. For the PSD cone case, the lower memory consumption is a plus.

Weaknesses: Lack of other baselines. Several improvements to CG have been proposed, that work both faster and better. (see detailed comments).

Correctness: Claims are correct, but empirically, a few more comparisons are warranted.

Clarity: yes

Relation to Prior Work: could be improved.

Reproducibility: Yes

Additional Feedback: Overall, the paper is well written and easy to understand. My main concern is the lack of comparisons to baselines that improve over vanilla CG, especially those that use line search, and other heuristics to improve performance [1], [2], [3]. These methods, while having the same O(1/k) convergence rate seem to converge much faster in practice. [2] consider matrix completion, and phase retrieval style applications as well. Line 91 : please clarify definition of dist* _x0000_ Line 116 and line 127: I wonder if other variants that look at removing atoms and solving subproblems [1], [2] can be considered as well in this setting. [1]Ochs, Peter, and Yura Malitsky. "Model Function Based Conditional Gradient Method with Armijo-like Line Search." International Conference on Machine Learning. 2019. [2] Rao, Nikhil, Parikshit Shah, and Stephen Wright. "Forward–backward greedy algorithms for atomic norm regularization." IEEE Transactions on Signal Processing 63.21 (2015): 5798-5811._x0000_ [3] G. Braun, S. Pokutta, and D. Zink. Lazifying conditional gradient algorithms. Proceedings of ICML, 2017. alg 2: line 11: if qq’ has to be computed, this still means you need O(n^2) space right? how is this different from prior CG methods where again you only need to store rank1 components? I suppose the key is you dont need to compute Xk. Edit: I read the other reviews and the author response. I'm still not fully sure about the superiority of the method, but my scalability concern has been addressed. Adding a 1 to my score due to that.

[Author Response · NeurIPS 2020]

## Author Response

We thank the reviewers for their valuable feedback.

**Experiment comparing CD and CG**   *R2*: You make a great point that each iteration of CD uses two line searches whereas CG uses only one. For problems in which the cost of a line search is comparable to that of solving the subproblem, the iteration count is probably not the most fair metric with which to compare CD and CG. Because of this, in Figure 1, we compare the number of matrix multiplications used by CD and CG instead. At each iteration of CD and CG, the computational cost of line searches is dwarfed by that of the repeated matrix multiplications with $\mathcal{G}^*(\nabla f(\eta_k y_k + (\eta_k - 1)g))$ necessary to compute the minimum eigenvalue via the Lanczos algorithm. If it were the case that the additional line search at each iteration of CD were only saving one minimum eigenvalue computation, then we would expect that CD would use about half as many matrix multiplications as CG. However, the histogram indicates that CD empirically does better. We cannot prove why this occurs, but we speculate on line 230.

**Experiment comparing CD and BM**   *R3*: The large dots on the left in Figure 2 show the performance of a traditional BM implementation at various ranks. We will annotate this directly on the plot to make it more clear. We see that pure BM, *i.e.*, without the safeguard, does not converge to the global optimum for $r = 2, 3, 4$ or 5. Even without the greedy heuristic, CD outperforms pure BM for $r = 2, 3, 4$, albeit after more matrix multiplications. At a high enough rank, pure BM will probably converge to the optimal value, but we are primarily focused on problems for which memory is the limiting factor.

**Assumption of no nonzero direction of recession**   *R3*: We agree that the assumption that $f$ have no nonzero direction of recession in $K$ is cludgy. To guarantee this is satisfied in problem (8), we could add an extremely small trace penalty. This probably would not affect our numerical results in any way. As an alternative assumption, we could stipulate that the set of optimal points is bounded and nonempty. This would imply that there is no nonzero direction of recession of $f$ in $K$.

**Matrix sketching and memory efficiency**   *R1*: We feel that the theory and practical utility of randomized matrix sketches in an optimization algorithm have been well established already in the literature, *e.g.*, [10, 20, 22, 23], and due to the space limitations, we treat the sketches essentially as tools. We will improve the citations to the relevant literature in Section 4. In Appendix F, we will move the citations on line 453 directly to line 452. *R2*: The use of the sketch means that we do not retain the $q_k$ at each iteration. The parameter in the sketch determines how much memory we use (only $3n$ numbers in the experiment in Figure 1). *R4*: In line 10 of Algorithm 2, we do not need to form $q_k q_k^T$. The operator $\mathcal{G}$ can be evaluated efficiently on rank-1 matrices because $\mathcal{G}(q_k q_k^T) = (q_k^T G_1 q_k, \ldots, q_k^T G_m q_k)$ and matrix vector products are efficiently computable for the $G_i$ (by assumption). The same idea holds for line 7 of Algorithm 5. In our implementation, we make sure to exploit this.

**Burer-Monteiro**   *R1*: It is fair to point out that CD is a heuristic. We should be clearer about that. The assumptions we mention in line 44 that are required in order to guarantee the global convergence of BM are very difficult to establish in practice. As far as we are aware, they are essentially the matrix version of the restricted isometry property (restricted strong convexity and restricted smoothness), and the rank necessary for the guarantees is frequently higher than that needed to specify the solution. Verifying the assumptions we make regarding problem (5) is relatively easy and can frequently be carried out by inspection. For memory-efficient CD, we really only need to check whether $G_i$ have efficient matrix multiplication routines and how large $m$ is relative to $n^2$.

**Related works**   *R1*: We do not address CGAL in this paper because we only consider the cone constraint. However, we think that directly handling equality constraints is the next natural step for future work. The connection between OMP and greedy conic optimization has been examined in detail in [16], and unfortunately, we do not think that we have the space to cover it here adequately. *R3*: Thank you for bringing the paper by Journee et al. to our attention. We will include it in our introduction. *R4*: Regarding the Ochs et al. paper, we do not see how to make meaningful comparisons for large scale optimization over the PSD cone using any model function other than the one which reduces to CG, *i.e.*, a simple minimum eigenvalue computation at each iteration. Regarding the Rao et al. paper, in order to maintain memory-efficiency for very large $n$, we cannot retain the prior iterates necessary to run CoGEnt. Regarding the Braun et al. paper, caching the prior iterates as part of the weak separation oracle would also significantly increase the potential memory requirements.

**Other**   *R1*: We will adjust our citations to reference published versions instead of ArXiv pre-prints whenever possible. Also, we will clean up the sentence on line 279. *R3*: We will add the citation to [22] in line 172. *R4*: The notation **dist**$^*$ is defined on lines 83-84.

[Meta-Review · NeurIPS 2020]

The consensus of the reviewers, after rebuttal and discussion, was to accept this paper for NeurIPS. The paper presents an interesting new algorithm that addresses some lingering issues in the application of the conditional gradient method when the constraint set is an unbounded cone. The contribution was decided to be worth publishing in NeurIPS, however, the paper can be improved along the lines suggested by the reviewers, especially to provide more evidence of the superiority of the algorithm to other known approaches. The authors are encouraged to address these comments.